# Bayesian Hyper-Parameter Optimisation for Malware Detection

Fahad T. ALGorain *,† and John A. Clark *,†

Department of Computer Science, University of Sheffield, Sheffield S10 2TN, UK
* Correspondence: ftalgorain1@sheffield.ac.uk (F.T.A.); john.clark@sheffield.ac.uk (J.A.C.)
† These authors contributed equally to this work.

**Abstract:** Malware detection is a major security concern and has been the subject of a great deal of research and development. Machine learning is a natural technology for addressing malware detection, and many researchers have investigated its use. However, the performance of machine learning algorithms often depends significantly on parametric choices, so the question arises as to what parameter choices are optimal. In this paper, we investigate how best to tune the parameters of machine learning algorithms—a process generally known as hyper-parameter optimisation—in the context of malware detection. We examine the effects of some simple (model-free) ways of parameter tuning together with a state-of-the-art Bayesian model-building approach. Our work is carried out using Ember, a major published malware benchmark dataset of Windows Portable Execution metadata samples, and a smaller dataset from kaggle.com (also comprising Windows Portable Execution metadata). We demonstrate that optimal parameter choices may differ significantly from default choices and argue that hyper-parameter optimisation should be adopted as a 'formal outer loop' in the research and development of malware detection systems. We also argue that doing so is essential for the development of the discipline since it facilitates a fair comparison of competing machine learning algorithms applied to the malware detection problem.

**Keywords:** hyper-parameter optimisation; automated machine learning; static malware detection; tree parzen estimators; bayesian optimisation; random search; grid search

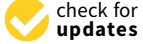



## 1. Introduction

Malware is one of the most pressing problems in modern cybersecurity, and its detection has been a longstanding focus for both academic and commercial research and development [1]. Detection must be effective (low false-positives and false-negatives) but also efficient, particularly in areas such as forensics or threat hunting, where vast file storage may need to be scanned for malware. Furthermore, the malware environment constantly changes, and so the (re-)training speed of detectors must also be considered [2]. Machine learning (ML) is an obvious avenue to pursue, with various advantages to harnessing it: an ML approach can significantly reduce manual effort in developing detectors, giving more rapid deployment; it can play a critical role in the extraction of insight from malware samples; and it can detect some unseen malware, e.g., unseen malware that has features that are similar to those of known malware may be detected because of the loose pattern matching that underpins ML classification approaches. A large number of ML techniques have been brought to bear on the malware detection problem, often attaining good results. However, ML must not be seen as a toolkit that can simply be thrown at a problem. Many ML techniques are parametrised, and the choice of parameters may make a significant difference to performance. In modern, widely used toolkits, ML algorithms often have many tens of parameters (and sometimes more). This raises the issue of how such parameters may be best set; a problem generally referred to as hyper-parameter optimisation (HPO). It is a significant focus of research in the optimisation community. Within malware detection, it has the potential to improve the results obtained by specific detection approaches that use default parameter choices and to enable a fair comparison of novel techniques with

existing techniques. (Comparing a novel technique with 'vanilla' or untuned variants of existing techniques is a recurring motif in the literature). The Authors of [3] advocate that hyper-parametrisation should be a 'formal outer loop' in the ML learning process, a view we very much support.

The size of the parameter space often means that manual tuning is practically impossible. ML toolkits seek to address this problem by adopting default values for parameters—values that have been shown to work plausibly well over many problems. However, for any specific problem at hand, it is far from clear that the default values will be the best, or even good, choices. Furthermore, we have significant domain incentives to gain the best possible results for malware detection. Both false-positive (FP) and false-negative (FN) classifications have major consequences. The former lead to significant wasted effort investigating the (non-) malware together with denial of service to the application concerned, while the latter means malware goes undetected, with potentially catastrophic consequences. For ML-based malware detection, we can conclude that finding high-performing hyper-parameters matters. In this paper, we explore the use of ML techniques applied to the classification of a specific form of malware: Windows Portable Execution (PE) files. We show that a specific technique is highly promising and that HPO still has a significant effect on its malware detection performance. We argue that HPO should play an important part in ML-based malware detection research and development,and in security applications more widely. The contributions of our paper are:

1. a demonstration of how well various ML-based Windows Portable Executable (PE) file classifiers perform when trained with default parameters.
2. an evaluation of various HPO approaches applied to this problem, including:

    (a) established major model-free techniques (Grid Search and Random Search); and
    (b) a state-of-the-art Bayesian optimisation model-based approach (Bayesian Optimisation with Tree-Structured Parzen Estimators).

3. a demonstration for our target problem that the optimal choices of ML hyper-parameters may vary considerably from the toolkit defaults.

Windows PE files are an important malware vector, and their detection has been the focus of significant research. The work described in this paper primarily uses the Ember dataset [4]—a recently published dataset comprising a header and derived information from a million PE files. The dataset contains samples of malware, benign software, and software of unknown status. These samples are labelled accordingly. Ember is now a major resource for the research community. We augment our Ember-focused work with work on a smaller PE dataset available from the high-profile competition website kaggle.com. In Section 2, we highlight previous work related to ML-based Windows PE malware detection and discuss relevant HPO literature. Section 3 provides the motivation for our work, defines the HPO problem and describes some major HPO techniques. Section 4 details the experiments performed. The results are given in Section 5, and Section 6 discusses the limitations of the work and indicates future work. Section 7 provides conclusions.

## 2. Related Literature

Numerous works have explored the use of machine learning for Windows PE malware detection, e.g., [5–7], but work has often been hampered by the absence of a standard benchmark dataset. The publication of the Ember dataset [4] has resolved this problem. The dataset is accompanied by various Python routines to facilitate access. Ember's authors have also provided baseline applications of various ML techniques to their dataset. In [8], the authors considered imbalanced dataset issues and model training duration. They also applied a static detection method using a Gradient-Boosting Decision Tree Algorithm. Their model achieved better performance than the baseline model with less training time. (They used feature reduction based on the recommendation of the authors of [4]). Another approach used a subset of the Ember dataset for their work and compared different ML models [9]. Their work is mainly concerned with scalability and efficiency. Their goal was

to identify malware families. The proposed Random Forest model achieved a slightly better performance than the baseline model.

Multiple works in the general optimisation literature have demonstrated the potential of HPO. For example, Reference [10] indicated the importance of parameter tuning for increasing accuracy, indicating that Random Search works better than Grid Search when tuning neural networks. Further, Reference [11] applied standard tuning techniques to a decision tree on 102 datasets and calculated the accuracy differences between tuned and traditional models. For all datasets, the experiments showed that tuning could achieve better performance than with the defaults. References [12,13] are concerned with greedy forward search, which seeks to identify the most important hyper-parameter to change next. Reference [14] stressed the importance of single hyper-parameters after using sequential model-based optimisation (SMBO) tuning. ANOVA was used to measure hyper-parameter importance. The authors of [15,16] assessed the performance of hyper-parameters across different datasets. Both have highlighted the importance of knowing which parameters to include in the hyper-parameter search space in order to see an improvement. Reference [16] also used surrogate models that allow setting randomly chosen hyper-parameter configurations based on a limit on the number of evaluations carried out. A hyper-parameter search based on Bayesian Optimisation (BO) was used in [15,17] to improve the speed of the search. The literature reveals that HPO, and in particular BO approaches, have much to offer. Readers are encouraged to refer to the survey paper [18] for a wider assessment of different HPO methods.

HPO has clearly given excellent results across many parameter optimisation problems. Below, in Section 3, we provide a more formal definition of HPO, consider in more detail its application to malware detection, and describe major HPO approaches.

## 3. Hyper-Parameter Optimisation

### 3.1. Formal Definition of HPO and Motivation for Its Use in Malware Classification

Hyper-parameters are parameters of a model that are not updated during the learning process [17]. The HPO problem is defined in a common way by many researchers as a search to find $x^*$ defined in Equation (1).

$$x^* = \arg\min_{x \in X} f(x), \tag{1}$$

where $f(x)$ is an objective function. Commonly, $f(x)$ is an error rate of some form evaluated on the validation set, e.g., the Root Mean Square Error (RMSE). $x^*$ is the hyper-parameter vector that gives rise to the lowest objective score, and $x$ can be any vector of parameters in the specified domain. HPO seeks the hyper-parameter values that return the lowest score. For malware and similar classification tasks, suitable choices for the objective functions are holdout and cross-validation error. Furthermore, if we consider a loss function for the same problem, then a possible choice is the misclassification rate [19]. For our proposed model, the loss function is defined by Equation (2).

$$f(x) = (ROC\_AUC - 1) \tag{2}$$

where $ROC\_AUC$ is the Receiver Operating Characteristic (with cross-validation) Area Under the Curve. ($ROC\_AUC$-related criteria are common in malware detection). For an in-depth background about validation protocols see [20]. Our work also aims to investigate evaluation time. There are two clear ways to do this. The first is to use a subset of folds in testing an ML algorithm [21]. The second is to use a subset of the dataset, especially if the data set is large [22,23], or to use fewer iterations.

Although HPO has a great deal to offer, it comes at a computational price. For every hyper-parameter evaluation, we must train the model, make predictions on the validation set, and then calculate the validation metrics. Developing a robust ML-based classifier for Windows PE with a credibly sized and diverse dataset such as Ember is, therefore, a significant undertaking. The computational costs involved act as a disincentive to im-

plementing Bergstra et al.'s formal outer loop. There is a pressing need for traversing the hyper-parameter space efficiently, and we demonstrate how a leading HPO approach allows us to do so.

Here, Windows PE files are a means to an end; the same issues apply to detecting other malware. Although malware is our major interest, our work also seeks to motivate consideration of HPO, and the use of state-of-the-art approaches, in particular, more widely in the application of ML in cybersecurity. For more information about HPO, interested readers should refer to [19].

### 3.2. Model-Free Blackbox Optimisation Methods

Perhaps the two most common HPO methods are Random Search and Grid Search. These require only an evaluation function to work, i.e., they are what is commonly referred to as 'blackbox' techniques.

Random Search selects values randomly from the domain of each hyper-parameter. Usually, the values selected from different domains by Random Search are independent, i.e., the value selected for one parameter does not affect the value selected for a different parameter. Furthermore, for an individual parameter, all values have the same probability of being selected. (Selection is said to be uniform). It is possible to relax such properties, producing what is referred to as a biased stochastic search. Such bias often encodes for domain insight, which is not in the spirit of a blackbox approach. In our work, we adopted a standard unbiased Random Search.

In Grid Search the individual parameters are discretised, i.e., a number of specific values are selected as 'covering' the particular parameter space. For example, the elements in the set $\{0.0, 0.25, 0.5, 0.75, 1.0\}$ could be taken to cover a continuous parameter in the range [0.0, 1.0]. Grid Search evaluates the function over the cross-product of the discretised hyper-parameter domains and so suffers from the 'curse of dimensionality' [24]. As the number of parameters increases or finer grain discretisation is adopted, the computational complexity mushrooms.

Random Search and Grid Search do not learn from past evaluations; we generally refer to such approaches as being *uninformed*. Consequently, they may spend a great deal of time evaluating candidates in regions where the previous evaluation of candidates has given rise to poor objective values. Random Search will search the specified space until a certain number of evaluations, time, or budget has been reached. It works better than Grid Search when we know the promising hyper-parameter regions, and so we can constrain the stochastic selection of candidates to lie in such regions [10,25]. Combining Random Search with complex strategies allows a minimum convergence rate and adds exploration that can improve model-based searches [19,26].

It is not surprising that uninformed methods can be outperformed by methods that use evaluation history to judge where to try next; indeed, such guided searches usually outperform uninformed methods [15,27,28].

### 3.3. Bayesian Optimisation (BO)

BO has emerged recently as one of the most promising optimisation methods for expensive blackbox functions. It has gained a lot of traction in the HPO community, with significant results in areas such as image classification, speech recognition, and neural language modelling. For an in-depth preview of BO, the reader is referred to [17,29]. BO is an informed method that takes into consideration past results to find the best hyper-parameters. It uses those previous results to form a probabilistic model that is based on a probability of the score given a vector of hyper-parameters. This is denoted by the formula: $P(score|hyperparameter)$. Reference [30] refers to the probabilistic model as a *surrogate* for the objective function denoted by $P(y|x)$, the probability of $y$ given $x$. The model or surrogate is more straightforward to optimise than the objective function. BO works to find the next hyper-parameters to be evaluated using the actual objective function by selecting the best-performing hyper-parameters on the surrogate function.

A five-step process to do this is given by [30]. The first step builds a surrogate probability model of the objective function. The second finds the hyper-parameters with the best results on the surrogate. The third applies those values to the real objective function. The fourth updates the surrogate using this new real objective function result. Steps 2–4 are repeated until the maximum iteration or budgeted time is reached [31]. BO has two primary components: a probabilistic model and an acquisition function to decide the next place to evaluate. Furthermore, BO trades off exploration and exploitation; instead of assessing the costly blackbox function, the acquisition function is cheaply computed and optimised. There are many choices for the acquisition function, but here, we use the most common—expected improvement (EI) [32]. The goal of Bayesian reasoning is to become more accurate as more performance data is acquired. The previous five-step processes are repeated to keep the surrogate model updated after each evaluation of the objective function [15]. BO spends a little more time generating sets of hyper-parameter choices that are likely to provide real improvements whilst keeping calls to the actual objective function as low as possible. Practically, the time spent on choosing the next hyper-parameters to evaluate is often trivial compared to the time spent on the (real) objective function evaluation. BO can find better hyper-parameters than Random Search in fewer iterations [27]. In this paper, we investigate whether AHBO-TPE, a specific variant of BO, can, for Windows PE file malware detection purposes, find better hyper-parameters than Random Search and with fewer iterations.

*3.4. Sequential Model-Based Optimisation (SMBO)*

There are several options for the SMBO's evaluation of the surrogate model $P(y|x)$ [15]. One of the choices is to use Expected Improvement (EI)m, defined in Equation (3).

$$EI_{y^*}(x) = \int_{-\infty}^{y^*} (y^* - y)P(y|x)dy \tag{3}$$

here $y^*$ is the threshold value of the objective function, $x$ is the vector of hyper-parameters, $y$ is the actual value of the objective function using the hyper-parameters $x$, and $P(y|x)$ is the surrogate probability model expressing the probability (density) of $y$ given $x$. The goal is to find the best hyper-parameters under function $P(y|x)$. The threshold value $y^*$ is the best objective value obtained so far. We aim to improve (i.e., get a lower value than) the best value obtained so far. For such minimisation problems, if a value $y$ is greater than the threshold value, then it is not an improvement. Only values less than the threshold are improvements. For a value $y$ less than the threshold $y^*$, the improvement is $(y^* - y)$. By integrating over all such improvements, weighted by the density function, $P(y|x)$ gives the overall expected improvement given the vector of hyper-parameter values $x$. When better values of $x$ are found (i.e., giving rise to actual improvements in the real objective function), the threshold value $y^*$ is updated. The above description is an *idealised* view of Expected Improvement. In practice, the choice of threshold value is more flexible, i.e., $y^*$ need not be the best objective value witnessed so far; this is actually the case for the Tree-Parzen Estimator approach outlined immediately below.

*3.5. Tree-Structured Parzen Estimators (TPE)*

The Tree-Structured Parzen Estimators approach constructs its model using Bayesian rules. Its model $P(y|x)$ is built from two model components, as shown in Equation (4). One component, $l(x)$, models values less than a threshold and the other, $g(x)$, models values greater than that threshold.

$$P(x|y) = \begin{cases} l(x) & if \ \ y < y^* \\ g(x) & if \ \ y >= y^* \end{cases} \tag{4}$$

TPE uses $y^*$ to be some quantile $\gamma$ of the observed $y$ values, i.e., such that $P(y < y^*) = \gamma$ [3]. This allows data to be available to construct the indicated densities. $l(x)$ is the density based

on the set of evaluated values of $x$ that have been found to give objective values less than the threshold. $g(x)$ is the density based on the remaining evaluated $x$ values. Here, $P(x|y)$ is the density of hyper-parameter $x$ given an objective function score of $y$. Following [15] it is expressed as shown in Equation (5).

$$P(y|x) = \frac{P(x|y) * P(y)}{P(x)} \tag{5}$$

Reference [15] also show that to maximise improvement, we should seek parameters $x$ with high probability under $l(x)$ and low probability under $g(x)$. Thus, they seek to maximise $g(x)/l(x)$. The best such $x$ outcome is then evaluated in the actual objective function and will be expected to have a better value. The surrogate model estimates the objective function; if the hyper-parameter that is selected does not make an improvement, the model will not be updated. The updates are based upon previous history/trials of the objective function evaluation. As mentioned before, the previous trials are stored in (score, hyper-parameters) pairs by the algorithm after building the lower threshold density $l(x)$ and higher threshold density $g(x)$. It uses the history of these previous trials to improve the objective function with each iteration. The motivation to use TPE with SMBO to reduce time and find better hyper-parameters came from leading HPO papers [15,18,27,33]. SMBO uses Hyperopt [3]—a Python library that implements BO or SMBO. Hyperopt makes SMBO an interchangeable component that could be applied to any search problem. Hyperopt supports more algorithms, but TPE is the focus of our work. Our contribution lies in the demonstration of the usefulness of SMBO using TPE for malware classification purposes.

## 4. Experiments

Here we outline the experiments carried out and provide sample data and execution environment details. Discussion of the results is given in Section 5.

### 4.1. Execution Environment and Dataset

Our work uses two powerful toolkits: Scikit-learn [34] and Hyperopt [3]. The experiments were carried out using the Windows 10 operating system, with 8GB RAM, AMD Ryzen 5 3550 H with Radeon Vega Mobile Gfc 2.10 GHz, 64-bit operating system, and an x64-based processor. Further, we used a MacBook Air (running Catalina version 10.15), 1.8 GHz Dual-core Intel i5, 8 GB 1600 MHz DDR3, Intel HD graphics 6000 1536 MB. Version 2018 of the Ember dataset [35] was used. This dataset comprises 1 M labelled samples. We used 300 k benign and 300 k malicious samples for training, with 100 k benign and 100 k malicious samples for testing purposes. The 200 k unlabelled examples of the dataset were not used in our experiments. Our work concerns supervised learning only. The work also uses a second dataset built by [36] using PE files from [37]. The dataset has 19,611 labelled malicious and benign samples from different repositories (such as VirusShare). Its samples have 75 features. It is split into 80% training and 20% testing and can be found in [36]. All results were obtained using Jupyter Notebook version 6.1.0 and Python version 3.6.0. Furthermore, implementation details of our experiments can be found on our github repository [38].

### 4.2. Experiments with Default Settings

Table 1 shows the results when various ML techniques are applied with default parameter settings. The techniques include well-established approaches: Stochastic Gradient Descent classifier (SGD), Logistic Regression classifier (LR), Gaussian Naïve Bayes (GNB), K-nearest Neighbour (KNN), and Random Forest (RF) [34,39]. A state-of-the-art approach— LightGBM [40]—is also used. LightGBM has over a hundred parameters, and so introduces major challenges for hyper-parametrisation. Some of its categorical parameters (e.g., boosting type) give rise to conditional parameters. For initial experiments, we adopted the default parameter settings adopted by the Scikit-Learn toolkit for all techniques other than LightGBM (which has its own defaults). The evaluation metric is Area Under the Receiver Operating Characteristic Curve (ROC AUC) [41]. ROC AUC plays an important role in many secu-

rity classification tasks, e.g., it also occurs frequently as an evaluation metric in intrusion detection research.

**Table 1.** Score Comparison of ML Models with Default Parameters (Ember Dataset).

| ML Model | Time to Train | Score (AUC-ROC) |
|---|---|---|
| GNB | 11 min 56 s | 0.406 |
| SGD | 11 min 56 s | 0.563 |
| LightGBM Benchmark | 26 min | 0.922 |
| RF | 57 min and 52 s | 0.90 |
| LR | 1 h and 44 min | 0.598 |
| KNN | 3 h 14 min 59 s | 0.745 |

*4.3. Model Hyper-Parameter Optimisation*

The most promising of the evaluated ML algorithms, taking into account functional performance and speed of training, was LightGBM. We choose to further explore hyper-parameter optimisation on this technique. Since LightGBM has over 100 parameters, some of which are continuous, we simply cannot do an exhaustive search. Accordingly, we have had to select parameters as a focus in this work. We focused on what we believe are the most important parameters. For Grid Search, we had to be particularly selective in what we optimised. Moreover, for Random Search, we specified a budget of 100 iterations. We examine Grid Search, Random Search, and AHBO-TPE as HPO approaches. We, therefore, compare model-free (blackbox) approaches (Grid Search and Random Search), = and AHBO-TPE, an approach that uses evaluation experience to continually update its model and suggest the next values of the hyper-parameters. We applied AHBO-TPE in two phases, the first one we initially set to 3 iterations, while the second was allowed 100 more iterations for fair comparison (with Random Search).

**5. Results**

The 2018 version of Ember was developed to include samples that present challenges to ML classification approaches [35]. It can, therefore, present an excellent means to stress-test available ML-based malware detection approaches. Table 1 shows the result of applying a variety of ML approaches, instantiated with their corresponding default parameters, to classify the samples of this dataset. All results were obtained under the MacBook Air environment described in Section 4. Table 1 also shows that the various ML techniques vary hugely in their suitability for the classification of PE files. We can see that LightGBM is clearly the best performing approach, taking both time and score into account.

The subsequent tables summarise our attempts to apply HPO approaches to the most promising of the original ML techniques. Table 2 gives the results of applying a variety of HPO techniques. The LightGBM Benchmark results are those given in Table 1. Grid Search results were also obtained using the MacBook environment. The remaining results (AHBO-TPE and Random Search) were obtained using the Windows 10 laptop. The number of objective evaluations indicates the default number of evaluations of the approach for Light-GBM, the total number of evaluations of the Grid Search, and the index of the evaluation at which the best result was achieved for AHBO-TPE. Random Search and AHBO-TPE were allowed 100 evaluations. Grid Search required 965 evaluations. The ranges for parameters subject to variation are shown later in Table 3 (for Grid Search) and Table 4 (for Random Search). Random Search was allowed to explore a greater number of parameters and performs well. The meaningful application of Grid Search to this extended set of varied parameters would be computationally infeasible.

**Table 2.** Score Comparison of HPO Methods (Ember Dataset).

| Search Methods | Best ROC AUC Score | Number of Objective Evaluations | Time to Complete Search |
|---|---|---|---|
| Benchmark LightGBM Model | 0.922 | 100 | 26 min (MacBook) |
| Grid Search | 0.944 | 965 | Almost 3 months (MacBook) |
| Random Search | 0.955 | 60 | 15 days, 13 h and 12 min (Windows 10) |
| AHBO-TPE with 100 iterations (results after 3 iterations) | 0.957 (0.955) | 26 (3) | 27 days (4 h) (Windows 10) |

**Table 3.** LightGBM Grid Search Hyper-parameter Results (Ember Dataset).

| Hyper-Parameter | Grid Search Best Hyper-Parameter Settings | Range | Default Value |
|---|---|---|---|
| boosting_type | GBDT | GBDT, DART, GOSS | GBDT |
| num_iteration | 1000 | 500:1000 | 100 |
| learning_rate | 0.005 | 0.005:0.05 | 0.1 |
| num_leaves | 512 | 31:2048 | 31 |
| feature_fraction | 1.0 | 0.5:1.0 | 1.0 |
| bagging_fraction | 0.5 | 0.5:1.0 | 1.0 |
| objective | binary | binary | None |

**Table 4.** LightGBM Random Search Hyper-parameter Results (Ember Dataset).

| Hyper-Parameter | Random Search Best Hyper-Parameter Settings | Range | Default Value |
|---|---|---|---|
| boosting_type | GBDT | GBDT or GOSS | GBDT |
| num_iteration | 60 | 1:100 | 100 |
| learning_rate | 0.0122281 | 0.005:0.05 | 0.1 |
| num_leaves | 150 | 1:512 | 31 |
| feature_fraction | 0.8 | 0.5:1.0 | 1.0 |
| bagging_fraction | 0.8 | 0.5:1.0 | 1.0 |
| objective | binary | binary only | None |
| min_child_samples | 165 | 20:500 | 20 |
| reg_alpha | 0.102041 | 0.0:1.0 | 0.0 |
| reg_lambda | 0.632653 | 0.0:1.0 | 0.0 |
| colsample_bytree | 1.0 | 0.0:1.0 | 1.0 |
| subsample | 0.69697 | 0.5:1.0 | 1.0 |
| is_unbalance | True | True or False | False |

We can see that HPO can offer significant improvements. Random Search performs very well, and so does AHBO-TPE. We can see that the initial optimisation for AHBO-TPE is far more efficient, with the technique achieving 0.955 after only three objective evaluations. Note that the time to completion is for information only. The LightGBM and Grid Search are evaluated on a Mac, and the remaining approaches were evaluated on a laptop running Windows (as described earlier)

AHBO-TPE achieves a very good result very quickly, i.e., after 3 iterations. Figure 1 illustrates the best score values achieved by Random Search and AHBO-TPE for each iteration (up to 100).

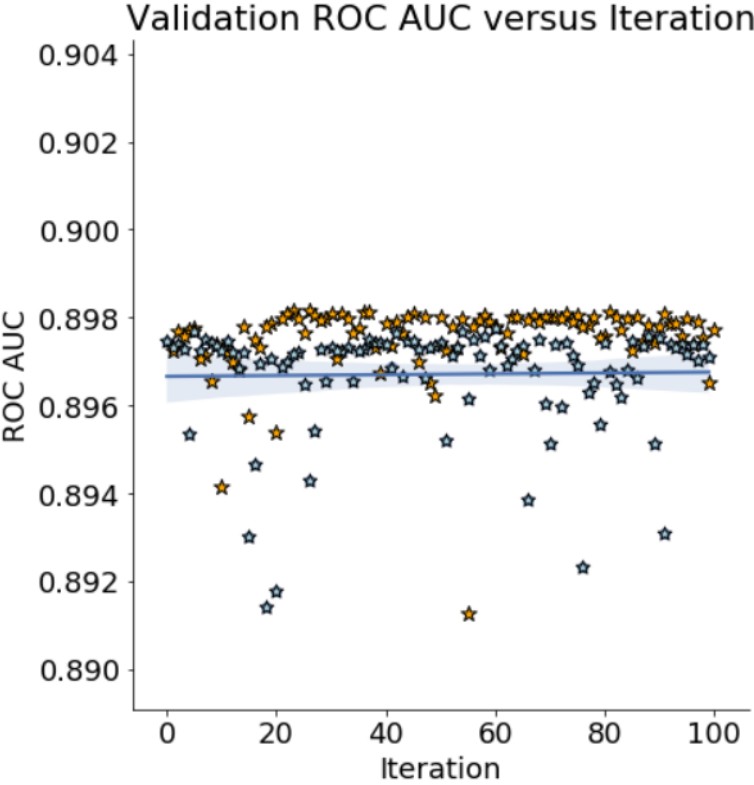

**Figure 1.** Highest Validation Score at each Iteration for AHBO-TPE (yellow) and Random Search (blue) (Ember Dataset).

Table 5 shows the performance of the remaining ML models using default parameter values and after parameter optimisation (using AHBO-TPE). All results were obtained under the Windows 10 environment indicated in Section 4.

**Table 5.** Score Comparison of the Remaining ML Models using AHBO-TPE (Ember Dataset).

| ML Model | Score (AUC-ROC) | Score (AUC-ROC) after Optimisation |
|---|---|---|
| GNB | 0.406 | same |
| SGD | 0.563 | 0.597 |
| RF | 0.901 | 0.936 |
| LR | 0.598 | 0.618 |
| KNN | 0.745 | 0.774 |

Table 6 gives the performance in training time based on the results attained by using AHBO-TPE. All results were obtained using the Windows 10 environment indicated in Section 4.

**Table 6.** Completion Time Results for Selected ML Models with AHBO-TPE (Ember Dataset).

| ML Model | Time to Train | Training Time Reduction |
|---|---|---|
| GNB | 11 min 56 s | same |
| SGD | 4 min 35 s | 14 min 35 s |
| LightGBM Benchmark | 18 min 30 s | 7 min 30 s |
| RF | 31 min 14 s | 26 min |
| LR | 1 h 5 min 37 s | 38 min |
| KNN | 4 h 37 min and 30 s | increased by 1 h 23 min 29 s |

Table 7 provides the default parameter results for various ML techniques applied to the kaggle.com dataset together with results after parameter optimisation using AHBO-

TPE. (For LightGBM, it also gives results where Random Search and Grid Search were used to optimise parameters). The LightGBM and RF Classifiers performed comparably, giving the highest AUC ROC scores (0.97914 and 0.97965). GNB and LR classifiers performed worst (0.54479 and 0.5072). The KNN classifier performs well (0.9595). SGD achieved a reasonable score (0.8432). The increase in score under AHBO-TPE for LightGBM is considerable (0.97914 to 0.99755). The tool's default parameter choices cannot be relied upon to produce the best or even good results.

**Table 7.** Score comparisons for the Application of HPO (Kaggle Dataset).

| ML Model | Default AUC ROC Score | Grid Search Optimised AUC ROC Score | Random Search AUC ROC Score | AHBO-TPE AUC ROC Score |
|---|---|---|---|---|
| LightGBM | 0.97914 | 0.98247 | 0.99809 | 0.99755 |
| RF | 0.97965 | N/A | N/A | 0.97819 |
| KNN | 0.94888 | N/A | N/A | 0.95954 |
| LR | 0.5 | N/A | N/A | 0.50729 |
| SGD | 0.84065 | N/A | N/A | 0.84322 |
| * GNB | 0.54475 | N/A | N/A | Same |

* There are no hyper-parameters for GNB; hence AHBO-TPE results are the same value as for defaults.

Tables 3–8 illustrate the difficulty of manually tuning parameters. In some cases, the defaults and the best-found values are at the opposite ends of the parameter ranges, e.g., the bagging fraction in Table 3. Many are significantly different from the default value, e.g., *num_leaves* in Tables 4 and 8 and *n_estimators* in Table 8. Some binary choices are reversed, e.g., *objective* and *is_unbalanced* of Table 4.

The hyper-parameters giving the best performance for each ML model are given in Tables 8–12. Here, AHBO-TPE was used as the HPO approach. The results are shown with 10 iterations (a constraint imposed for reasons of computational practicality) and 3-fold cross-validation. All results were obtained using the Windows 10 environment indicated in Section 4.

**Table 8.** LightGBM AHBO-TPE Search Hyper-parameter Results (Ember Dataset).

| Hyper-Parameter | Random Search Best Hyper-Parameter Settings | Range | Default Value |
|---|---|---|---|
| boosting_type | GBDT | GBDT or GOSS | GBDT |
| num_iteration | 26 | 1:100 | 100 |
| learning_rate | 0.02469 | 0.005:0.05 | 0.1 |
| num_leaves | 229 | 1:512 | 31 |
| feature_fraction | 0.78007 | 0.5:1.0 | 1.0 |
| bagging_fraction | 0.93541 | 0.5:1.0 | 1.0 |
| objective | binary | binary only | None |
| min_child_samples | 145 | 20:500 | 20 |
| reg_alpha | 0.98803 | 0.0:1.0 | 0.0 |
| reg_lambda | 0.45169 | 0.0:1.0 | 0.0 |
| colsample_bytree | 0.89595 | 0.0:1.0 | 1.0 |
| subsample | 0.63005 | 0.0:1.0 | 1.0 |
| is_unbalance | True | True or False | False |
| n_estimators | 1227 | 1:2000 | 100 |
| Subsample_for_bin | 160,000 | 2000:200,000 | 200,000 |

**Table 9.** SGD Model AHBO-TPE Search Hyper-parameter Results (Ember Dataset).

| Hyper-Parameter | AHBO-TPE Search Hyper-Parameter Results | Range | Default Value |
|---|---|---|---|
| Penalty | L2 | L1, L2, elasticnet | L1 |
| Loss | Hinge | hinge, log, modified-huber, squared-hinge | Hinge |
| Max-iterations | 10 | 10:200 | 1000 |

**Table 10.** RF Model AHBO-TPE Search Hyper-parameter Results (Ember Dataset).

| Hyper-Parameter | AHBO-TPE Search Hyper-Parameter Results | Range | Default Value |
|---|---|---|---|
| n_estimators | 100 | 10:100 | 10 |
| max_depth | 30 | 2:60 | None |
| max_features | auto | auto, log2, sqrt | auto |
| min_samples_split | 10 | 2:10 | 2 |
| min_samples_leaf | 30 | 1:10 | 1 |
| criterion | gini | gini, entropy | gini |

**Table 11.** LR Model AHBO-TPE Search Hyper-parameter Results (Ember Dataset).

| Hyper-Parameter | AHBO-TPE Search Hyper-Parameter Results | Range | Default Value |
|---|---|---|---|
| max_iter | 200 | 10:200 | 100 |
| C | 8.0 | 0.0:20.0 | auto |
| solver | sag | liblinear, lbfgs, sag, saga | lbfgs |

**Table 12.** KNN Model AHBO-TPE Search Hyper-parameter Results (Ember Dataset).

| Hyper-Parameter | AHBO-TPE Search Hyper-Parameter Results | Range | Default Value |
|---|---|---|---|
| n_neighbors | 15 | 1:31 | 5 |

In Figure 2, a comparison is given between the benchmark model results and those obtained using AHBO-TPE and Random Search to optimise parameters.

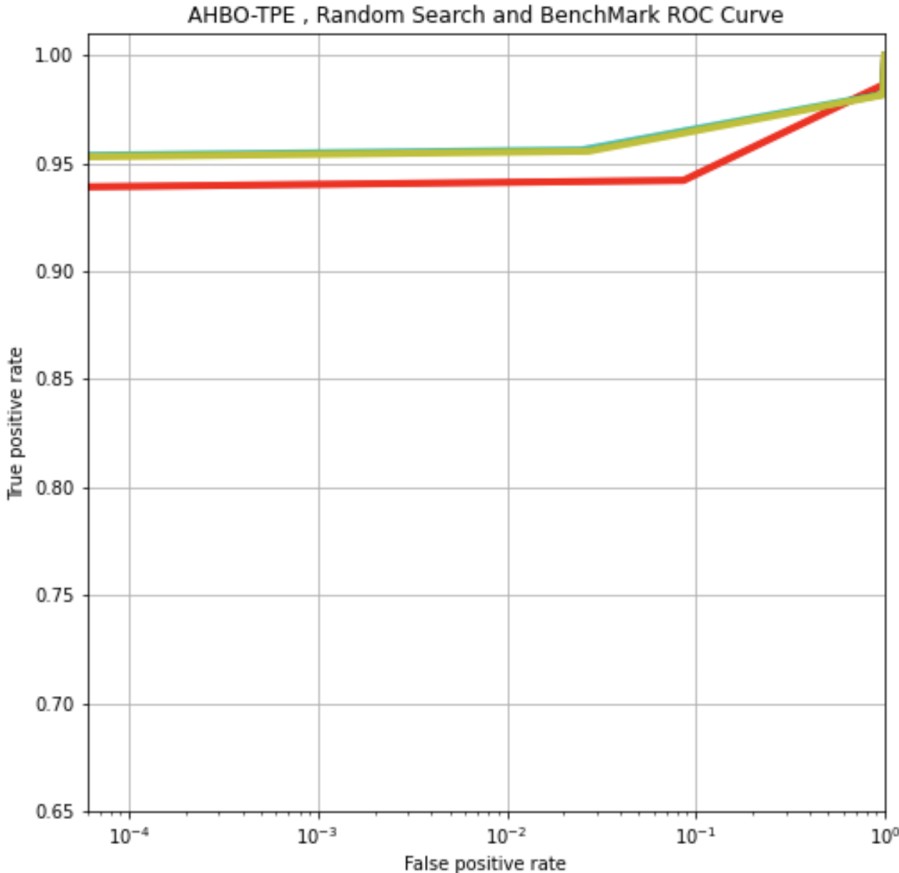

**Figure 2.** ROC AUC Comparison for AHBO-TPE (Cyan), Random Search (Yellow), and Default Benchmark Model (Red) applied to the Ember Dataset.

Moreover, Table 13 illustrates the highest performing parameters,obtained using AHBO-TPE for the kaggle.com dataset.

**Table 13.** ML Models Hyper-parameter results using AHBO-TPE (Kaggle Dataset).

| ML Models | Hyper-Parameter | Range | Best Hyper-Parameter Results |
|---|---|---|---|
| LightGBM | num_leaves | 1:512 | 10 |
| | Min_child samples | 20:500 | 90 |
| | n_estimators | 10:100 | 19 |
| | boosting_type | gbdt | Gbdt |
| | learning_rate | 0.01:0.5 | 0.4418140187193226 |
| | Subsample_for_bin | 2000:200,000 | 80,000 |
| | Colsample_bytree | 0.6:1.0 | 0.7307181013749854 |
| | feature_fraction | 0.5:1.0 | 0.6726481091942302 |
| | Bagging_fraction | 0.5:1.0 | 0.5893616201923844 |
| | Reg_alpha | 0.0:1.0 | 0.195989486417426 |
| | Reg_lambda | 0.0:1.0 | 0.1939778453324642 |
| | Is_unbalance | True, False | False |
| | objective | Binary | Binary |
| RF | n_estimators | 10:200 | 100 |
| | max_depth | 10:50 | 15 |
| | max_features | auto, sqrt | sqrt |
| | min_samples_split | 10:50 | 19 |
| | min_samples_leaf | 10:50 | 10 |
| | criterion | entropy, gini | entropy |
| KNN | n_neighbors | 1:100 | 3 |
| GNB | N/A | N/A | N/A |
| SGD | penalty | none, l1, l2, elasticnet | L2 |
| | loss | hinge, log, modified_huber, squared_hinge, perceptron | log |
| | max_iter | 20:1000 | 790 |
| | alpha | 0.0001:0.2 | 0.0001 |
| LR | Max_iter | 20:500 | 155 |
| | C | 1.0:50.0 | 7 |
| | solver | lbfgs, sag, saga | sag |

## 6. Discussion

The results show how the default values of parameters generally give suboptimal results and how optimal choices of the parameter values for various models can vary significantly from their defaults. The results also show that applying HPO to malware detection can be computationally practical. Where there are a great number of hyper-parameters (for example, LightGBM has more than one hundred), some *efficient* automated means of determining effective choices are essential. Credible manual tuning will not be feasible, and many HPO approaches may be computationally impractical.

The work has shown the utility of using proxy evaluation functions for determining hyper-parameter values. In particular, AHBO-TPE has been shown to be a very effective and efficient informed approach. Other forms of surrogates may bring benefits. For example, a deep neural network could be used as a function approximator (learned from real objective function evaluations). Such an approximator could be used in place of the computationally intensive real objective function in almost any search-based approach. The search could revert to using the real objective function starting from the best vector of hyper-parameters obtained by optimising using the neural network approximator.

We have used a single (albeit highly effective) 'informed' hyper-parametrisation approach. The use of other informed hyper-parametrisation approaches could provide further insight and possible improvements. For practical purposes, we informally identified plausible parameters that should be subject to variation and allowed the remaining ones to be set at the defaults. It is possible that improvements in results could be obtained by allowing variation in the parameters that were fixed at their default values. It also suggests the possibility of adopting a sequential approach to optimising over the full range of parameters, i.e., once investigated parameters have been subject to variation and evaluation, they could be fixed at their optimal values and previously fixed parameters then be allowed to vary. The focus of our work has been Windows PE files. Similar investigations of other malware types are now needed to determine how well our approach generalises.

## 7. Conclusions

We have shown that HPO matters a great deal for ML-based malware detection. The use of default parameters will generally not be optimal, and the results overall would suggest researchers in malware and ML are missing a significant opportunity to use HPO to improve results attained by specific techniques of interest. Every improvement matters to the security of the protected systems and reduces costs in one form or another: getting the best out of malware detectors matters a great deal, and HPO has much to offer. We have also shown that a specific informed technique (AHBO-TPE) has particular potential for application to malware detection.

Using HPO to provide Bergstra et al.'s 'formal outer loop' should be normal practice to ensure any targeted technique is exploited fully. Adopting HPO in this way brings methodological benefits: for the development of the field, we need to be able to compare competing techniques at their best, and HPO can provide a principled and repeatable way to get the best (or close to it) from all competing techniques. We propose that HPO be an essential element of the ML process for malware detection applications, i.e., that Bergstra et al.'s 'formal outer loop' be adopted, and recommend further research into the use of HPO for tuning malware detectors.

**Author Contributions:** Supervision, J.A.C.; Writing—original draft, F.T.A.; Writing—review and editing, F.T.A. and J.A.C. All authors have read and agreed to the published version of the manuscript.

**Funding:** The APC was funded by NTNU's IDUN project.

**Data Availability Statement:** Ember Dataset can be found in (https://github.com/elastic/ember, accessed on 19 November 2021) and Kaggle dataset can be found in (https://www.kaggle.com/datasets/amauricio/pe-files-malwares, accessed on 11 October 2021).

**Conflicts of Interest:** The authors declare no conflict of interest.

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
