# Peer review of "Bayesian Hyper-Parameter Optimisation for Malware Detection"

_electronics, doi:10.3390/electronics11101640_

Round 1
Reviewer 1 Report
This paper studies the performance of a variety of machine learning algorithms for building malware classifiers and hyperparameter optimization through a series of experiments. This paper is written well. The following comments could be considered to improve the paper further.
- The details of the experiments are insufficient. It is necessary to represent more values of hyperparameters of your experiments. Without these details, readers cannot reproduce the experimental results.
- The figures of this paper are not clear in a high-resolution environment, for example, Figures 1. It is recommended to use vector graphics.
- The references should have the same type. For example, the location and format of publication year information should be consistent.
- Recently, the gradient activation function is used to improve the performance of deep learning, e.g., ``Activated gradients for deep neural networks`` in IEEE TNNLS. The authors are suggested to add some comments in the revised manuscript on this point.
- There are some grammatical errors. I recommend authors use a language editor to improve the quality of the article.
- The description of the proposed experiments in the article is not clear enough, please strengthen it.
Author Response
Response to Reviewer 1 Comments
Point 1: The details of the experiments are insufficient. It is necessary to represent more values of hyperparameters of your experiments. Without these details, readers cannot reproduce the experimental results.
Response 1: We have updated the descriptions and the presentation of the results of our experiments; the hyper-parameters that were chosen are indicated in the corresponding tables. Also, there is a git hub profile that includes the whole process should anyone wish to reproduce our experiments.
Point 2: The figures of this paper are not clear in a high-resolution environment, for example, Figures 1. It is recommended to use vector graphics.
Response 2: We tried to fix the screenshot quality for the figures as much as possible, but we don’t have any expertise in vector graphics. This is as good as we can make it currently.
Point 3: The references should have the same type. For example, the location and format of publication year information should be consistent
Response 3: We have fixed the reference style using the natbib package.
Point 4: Recently, the gradient activation function is used to improve the performance of deep learning, e.g., ``Activated gradients for deep neural networks`` in IEEE TNNLS. The authors are suggested to add some comments in the revised manuscript on this point
Response 4: We haven’t used neural networks in any of our experiments. It seems odd to give any further detail on this matter.
Point 5: There are some grammatical errors
Response 5: We have improved the presentation in many places in the paper.
Point 6: The description of the proposed experiments in the article is not clear enough, please strengthen it
Response 6: We hope this is now clear. Essentially our experiments either apply ML classification techniques to the samples of PE metadata. Seeds are included in the GitHub repository Jupyter notebook files. Anyone wishing to reproduce our experiments can run those notebooks on the indicated platforms.
Reviewer 2 Report
In this article, the authors investigate the use of a variety of Machine Learning algorithms for building malware classifiers and also how best to tune the parameters of those algorithms – generally known as hyper-parameter optimization. However, I will comment on some aspects that must be improved for this manuscript, the changes performed must be highlighted:
-The authors incorrectly use the acronyms, the correct form to write them as in line 22. This error must be corrected in all the acronyms that exist in the manuscript.
-Line 14 must be deleted.
-The article format must be specified for which journal it is sent, in this case it is for Electronics.
-It is not necessary to write certain phrases in bold.
-Section 2 must eliminate the Subsections.
-The authors do not present the a brief of Sections that will come at the end of Section 1 or 2.
-The authors before presenting a Subsection, must write an introduction.
-The authors must correctly cite the references, especially check line 217.
-Improve the quality of English in the scientific article, especially avoiding the use of Phrasal Verbs.
-Where did the authors obtain the dataset for the experiments?
-Tables must have a correct format and be larger.
-More explanation of the results presented in the experimentation is required.
-Distribute equally the objects on page 7 and 8 between text.
-Section 6 is only Conclusions, it must not be mixed with Limitations.
-The authors must present the limitations in another section.
-The authors do not present details of the experimentation, and much less of the scenarios in which they have performed it.
-The authors must improve the conclusions and add future works.
-The References Section must be not numbered and must be placed in its respective MDPI format.
-The Appendix Section must be added to the manuscript and not be separated from the article.
-References must be carefully reviewed since they do not have a correct format.
Author Response
Response to Reviewer 2 Comments
Point 1: The authors incorrectly use the acronyms, the correct form to write them as in line 22. This error must be corrected in all the acronyms that exist in the manuscript.
Response 1: We believe our use of abbreviations is entirely appropriate. Note that we do not use any acronyms as such. An acronym is something like NATO, formed from the initial letters of its constituent words, but pronounced as a word: “Nay-toe”. We make regular use of abbreviations, e.g. ML and HPO. This imposes a lesser burden on the reader than using expanded forms throughout.
Point 2: Line 14 must be deleted.
Response 2: Done
Point 3: The article format must be specified for which journal it is sent, in this case, it is for Electronics
Response 3: We have fixed that already and it shows now that the selected Journal is Electronics.
Point 4: It is not necessary to write certain phrases in bold
Response 4: We have removed all bold phrases to avoid any irritation.
Points 4,5,6: Section 2 must eliminate the Subsections, The authors do not present a brief of Sections that will come at the end of Section 1 or 2
Response 4,5,6: We deleted section 2 subsections and we have added after the introduction a paragraph showing the sections to come and a brief about each.
Point 7: The authors must correctly cite the references, especially check line 217
Response 7: We have fixed the references and citations accordingly
Point 8: Improve the quality of English in the scientific article, especially avoiding the use of Phrasal Verbs
Response 8: This is hard to interpret. There would appear to be very few phrasal verbs. We have updated the narrative throughout to enable a more convincing narrative flow. Some parts have been simplified.
Point 9: Where did the authors obtain the dataset for the experiments?
Response 9: From kaggle.com; the link for the dataset is included in the paper, experiments section reference number 36
Point 10: Tables must have a correct format and be larger
Response 10: The table and text inside were fixed and are now better
Point 11: More explanation of the results presented in the experimentation is required
Response 11: We have made changes to represent the results of the experiments. Also, a Github repository was created to allow reproducibility.
Point 12: Distribute equally the objects on pages 7 and 8 between text
Response 12: We have not made this change since efforts to do so consistently messed up various aspects of the presentation.
Point 13,14: Section 6 is only Conclusions, it must not be mixed with Limitations
Response 13,14: Limitations and Conclusions are now separate sections.
Point 15: The authors do not present details of the experimentation, and much less of the scenarios in which they have performed it
Response 15: We have made changes and clearly specified how we have made our work and which technique was utilized. Also, as mentioned before, section 4.4 indicates our GitHub repository.
Point 16: The authors must improve the conclusions and add future works
Response 16: We have made changes and we hope it's better now.
Point 17: The References Section must be not numbered and must be placed in its respective MDPI format
Response 17: We have made changes according to the MDPI format using BibTex.
Point 18: The Appendix Section must be added to the manuscript and not be separated from the article
Response 18: We have made changes accordingly.
Point 19: References must be carefully reviewed since they do not have a correct format
Response 19: We have reviewed the reference carefully and made sure it is in the correct BibTeX format.

Reviewer 3 Report
This work investigates investigate the use of a variety of ML algorithms for building malware classifiers and also how best to tune the parameters of those algorithms generally known as hyper parameter optimisation. The following revisions are required.
- In literature review, add 3 to five more relevant and latest techniques.
- Add Comparison table at the end of section 2 and compare with at least 10 to 15 techniques with appropriate parameters.
- Please make sure your paper has necessary language proof-reading.
Author Response
Response to Reviewer 3 Comments
Point 1: In the literature review, add 3 to five more relevant and latest techniques
Response 1: We have not done that because of reference number 33 they have made a great survey outlining almost all of the latest techniques . (we have pointed that out in the paper)
Point 2: Add Comparison table at the end of section 2 and compare with at least 10 to 15 techniques with appropriate parameters
Response 2: We have not done that because of reference number 33 they have made a great survey comparing and outlining every technique and its parameter. (we have pointed that out in the paper)
Point 3: Please make sure your paper has necessary language proof-reading
Response 3: We have made the necessary proofreading and hopefully it is better now.

Round 2
Reviewer 1 Report
The manuscript has been improved and can be considered for acceptance.
Author Response
Thank you very much for your time and consideration,
The Authors
Reviewer 2 Report
Thanks to the authors for performing some of the suggested changes. However, the changes as mentioned in the previous revision have not been made.
Author Response
Point 1: The authors incorrectly use the acronyms, the correct form to write them as in line 22. This error must be corrected in all the acronyms that exist in the manuscript.
Response 1: We have fixed that hopefully its better now.
Point 2: Line 14 must be deleted.
Response 2: Done
Point 3: The article format must be specified for which journal it is sent, in this case it is for Electronics
Response 3: We have fixed that already and it shows now the selected Journal is Electronics.
Point 4: It is not necessary to write certain phrases in bold
Response 4: Removed all bold phrases
Point 4,5,6: Section 2 must eliminate the Subsections, The authors do not present a brief of Sections that will come at the end of Section 1 or 2
Response 4,5,6: We deleted seciton 2 subsection and we have added after the introduction a prargraph showing the sections to come and a breif about each section.
Point 7: The authors must correctly cite the references, especially check line 217
Response 7: Fixed the refernce and citation accordingly
Point 8: Improve the quality of English in the scientific article, especially avoiding the use of Phrasal Verbs
Response 8: We hope that it is much better now
Point 9: Where did the authors obtain the dataset for the experiments?
Response 9: From kaggle.com ; the link for the dataset is included in the paper, experiments section reference number 36
Point 10: Tables must have a correct format and be larger
Response 10: Table and text inside were fixed and it now better
Point 11: More explanation of the results presented in the experimentation is required
Response 11: We have made changes to represent the results acquired by the experiments further more, a github account were created for repreducibility.
Point 12: Distribute equally the objects on page 7 and 8 between text
Response 12: We have tried but it started missing up a lot of space and text – we just put it back to how it was safer and cleaner this way.
Point 13,14: Section 6 is only Conclusions, it must not be mixed with Limitations
Response 13,14: We have fixed it and put both in separate sections.
Point 15: The authors do not present details of the experimentation, and much less of the scenarios in which they have performed it
Response 15: We have made changes and clearly specified how we have made our work and which technique was utilised. Also, as I mentiond before details of our experiments is in our github account in section 4.4 additional materials.
Point 16: The authors must improve the conclusions and add future works
Response 16: We have made changes and we hope its better now.
Point 17: The References Section must be not numbered and must be placed in its respective MDPI format
Response 17: We have made changes according to the MDPI format using BibTex.
Point 18: The Appendix Section must be added to the manuscript and not be separated from the article
Response 18: We have made changes accordingly.
Point 19: References must be carefully reviewed since they do not have a correct format
Response 19: We have reviewed the reference carefully and made sure it is in the correct bit tex format and directly taken from citation section in google scholar.

Round 3
Reviewer 2 Report
Thanks to the authors for performing some changes suggested. However, there are more issues that authors must fix, which I detail below:
-Tables 3 to 5 and 13 are not mentioned in the manuscript. Also, Tables 6 to 13 must not be in Appendix Section.
-Tables are not referenced in incremental sorted.
-The acronyms are incorrectly written. The correct form is such Authors wrote in line 253: "Gaussian Naïve Bayes (GNB)".
-Figure 2 must not be in the Appendix Section.
-Section 6 must be renamed to Discussion. -Subsection 4.4 is tiny to be a subsection. For this reason, authors must unite with the previous subsection. -Equations are not referenced in the manuscript.
-All object as "Section", "Figure", "Algorithm", "Table", and "Equation" must be written with the first letter in capital letter. This error must be fixed in the manuscript especially in line 77.
-There are some blank space in the pages 10 to 13.
Author Response
Point 1: Tables 3 to 5 and 13 are not mentioned in the manuscript. Also, Tables 6 to 13 must not be in Appendix Section.
Response 1: We have fixed the tables according to the suggestions.
Point 2: Tables are not referenced in incremental sorted
Response 2: Tables are now referenced in order.
Point 3: The acronyms are incorrectly written. The correct form is such Authors wrote in line 253: "Gaussian Naïve Bayes (GNB)".
Response 3: We have fixed the acronyms now as specified. Point 4: Figure 2 must not be in the Appendix Section.
Response 4: Figure 2 is now removed from the Appendix section
Point 5: Section 6 must be renamed to Discussion. -Subsection 4.4 is tiny to be a subsection. For this reason, authors must unite with the previous subsection. -Equations are not referenced in the manuscript.
Response 5: Secion 6 is now renamed to Discussio. And the subsesction 4.4 is moved merged with subsection 4.1. Execution Environment and Dataset.
Point 6: All object as "Section", "Figure", "Algorithm", "Table", and "Equation" must be written with the first letter in capital letter. This error must be fixed in the manuscript especially in line 77.
Response 6: All objects are now fixed in the manuscript according to the
suggestion.
Point 7: There are some blank space in the pages 10 to 13.
Response 7: blank spaces are now reduced to the best that we can do.

Round 4
Reviewer 2 Report
The authors of this manuscript have not made some the changes suggested by the reviewers, and have even added new errors, I detail them below:
-The objects (Tables, Figures, etc) are cited too far from where they must be, for example Table 1 cited on page 6 and appears on page 8.
-There are still problems with the acronyms.
-The results of the Tables must be further explained in the manuscript to better understand the reader.
Author Response
Point 1: The objects (Tables, Figures, etc) are cited too far from where they must be, for example Table 1 cited on page 6 and appears on page 8.
Response 1: We have fixed the tables and the figures according to the suggestions. We have also added text to clarify more. But largely the import of the results is generally easily understood from the text.
Point 2: There are still problems with the acronyms.
Response 2: Acronyms were double-checked accordingly. We do not believe that our use of abbreviations is problematic.
Point 3: The results of the Tables must be further explained in the manuscript to better understand the reader.
Response 3: We have updated the descriptions of experiments and added clarifying information to many of the tables and text.